# 12-Year Trends in Active School Transport across Four European Countries—Findings from the Health Behaviour in School-Aged Children (HBSC) Study

**DOI:** 10.3390/ijerph18042118

**Published:** 2021-02-22

**Authors:** Ellen Haug, Otto Robert Frans Smith, Jens Bucksch, Catherina Brindley, Jan Pavelka, Zdenek Hamrik, Joanna Inchley, Chris Roberts, Frida Kathrine Sofie Mathisen, Dagmar Sigmundová

**Affiliations:** 1Department of Health Promotion and Development, University of Bergen, 5020 Bergen, Norway; frida.mathisen@uib.no; 2Department of Teacher Education, NLA University College, 5012 Bergen, Norway; 3Department of Health Promotion, Norwegian Institute of Public Health, 5015 Bergen, Norway; robert.smith@fhi.no; 4Department of Prevention and Health Promotion, Faculty of Natural and Human Sciences, Heidelberg University of Education, 69120 Heidelberg, Germany; bucksch@ph-heidelberg.de (J.B.); brindley@ph-heidelberg.de (C.B.); 5Department of Recreation and Leisure Studies, Faculty of Physical Culture, Palacký University Olomouc, 77111 Olomouc, Czech Republic; jan.pavelka@upol.cz (J.P.); zdenek.hamrik@upol.cz (Z.H.); 6MRC/CSO Social and Public Health Sciences Unit, University of Glasgow, Glasgow G37 HR, UK; Joanna.Inchley@glasgow.ac.uk; 7Social Research and Information Devision, Welsh Government, Cardiff CF10 3NQ, UK; Chris.Roberts@gov.wales; 8Institute of Active Lifestyle, Faculty of Physical Culture, Palacký University Olomouc, 77111 Olomouc, Czech Republic; dagmar.sigmundova@gmail.com

**Keywords:** active school transport, trends, cross-national, HBSC, gender, age, SES

## Abstract

Active school transport (AST) is a source of daily physical activity uptake. However, AST seems to have decreased worldwide over recent decades. We aimed to examine recent trends in AST and associations with gender, age, family affluence, and time to school, using data from the Health Behaviour in School-Aged Children (HBSC) study collected in 2006, 2010, 2014, and 2018 in the Czech Republic, Norway, Scotland, and Wales. Data from 88,212 students (11, 13 and 15 years old) revealed stable patterns of AST from 2006 to 2018, apart from a decrease in the Czech Republic between 2006 and 2010. For survey waves combined, walking to and from school was most common in the Czech Republic (55%) and least common in Wales (30%). Cycling was only common in Norway (22%). AST differed by gender (Scotland and Wales), by age (Norway), and by family affluence (everywhere but Norway). In the Czech Republic, family affluence was associated with change over time in AST, and the effect of travel time on AST was stronger. The findings indicate that the decrease in AST could be levelling off in the countries considered here. Differential associations with sociodemographic factors and travel time should be considered in the development of strategies for AST.

## 1. Introduction

In recent years, active travel has become an integral part of international initiatives aimed at increasing levels of physical activity within the population [1,2,3,4]. Walking and cycling to school have gained considerable attention as sources of young peoples’ daily physical activity. More recently, there has also been an increased focus on active commuting as a sustainable form of transport that can reduce problems caused by motorised vehicles, with potentially significant economic benefits and public health impacts [3,5,6,7,8].

Reviews of the growing body of literature in the field strongly support a positive relationship between active school transport (AST) and levels of physical activity [9,10]. Positive associations have also been found between cycling to and from school and cardiovascular fitness [9,10,11,12]. A relationship between AST and body composition indicators is less clear [9,10]. Other potential co-benefits of AST relate to improved navigation and road safety abilities [13,14], better processing of the physical surroundings [15], higher activation (i.e., alertness and activity) during school hours [16], and the development of long-term physical activity and active transportation habits [17,18].

Despite the potential benefits, the number of children and youths that walk or cycle to school seems to have decreased worldwide in recent decades [19,20,21,22,23,24,25,26,27,28]. However, large cross-country variations are observed in the prevalence of AST and the magnitude of decline [21]. In the United States, the prevalence of AST among children dropped from 49 to 13% between 1969 and 2009 [25], whereas it declined from 44 to 21% among 10–14 year old children in Australia between 1971 and 2003 [24]. Findings from Europe display a more mixed picture, with generally higher proportions of AST being reported, but decreasing trends have also been observed in European studies [22,26,29,30]. Many of the existing studies on trends in AST span over several decades, covering a period when use of motorised vehicles increased dramatically. Furthermore, the studies are limited to data from individual countries preventing reliable cross-national comparisons. More recently, policy initiatives and national programs to promote active school commuting have been initiated in many countries [3,6,31,32,33], which may have affected schoolchildren’s travel behaviours.

Contemporary studies suggest that AST is associated with a wide range of factors, such as demographic (gender, ethnicity, age), family (parental education, household income, car ownership), social (individual and parental attitudes and concerns, social and cultural norms), environmental (school distance, safety, walkability, traffic calming, infrastructure, recreational facilities, centralization), and policy-related factors [6,25,29,34,35,36]. It is to be expected that some of the underlying drivers of AST will vary between and within countries [23]. However, a large body of research has identified the distance between home and school and time taken to travel to school as the strongest predictors of AST [6,14,29,36,37], and findings from several countries point towards an increase in the distance to school over time [20,27,29,38].

Gender, age, and socioeconomic status (SES) have been identified as potential moderators of AST [39]. Findings from North America, Australia, New Zealand, and the Czech Republic indicate higher levels of AST for boys more often than girls [14,32,36,40,41], while gender differences have not been observed in studies from Switzerland [26] and Norway [11]. Regarding gender difference trends in recent times, there is no clear pattern, with no differences [41,42], a decline only for girls [43] and a decline in boys and an increase in girls [27] having been observed. The relationship between age and AST is expected to be curvilinear, with an initial age-related increase due to more independent mobility and parental allowances, followed by a decrease because of generally longer distances to secondary schools compared with localised primary schools [14]. Socioeconomic differences in AST have been less well studied. The literature from North America and New Zealand generally shows that low-income households and lower parental education correlate with more AST [14,36]. Nevertheless, the lack of standardised measures and comparable control variables across different studies makes it difficult to compare, aggregate and interpret findings [35,36,44,45].

Cross-national studies of current time trends with a comparable methodological approach are of interest because they can provide unique insights into how recent developments, as well as national and local level policies, may have had an impact on AST. It has been suggested that future research should also consider changes in key AST correlates over time to support the development of new policies, regulations, designs, and programme interventions [36]. To improve the understanding of young peoples’ transport to school in different regions in Europe between 2006 and 2018, the current study aimed to examine secular trends in AST and their associations with gender, age, SES, and time to school across Northern Europe (Norway), western Europe (Scotland and Wales) and central Europe (the Czech Republic).

## 2. Materials and Methods

### 2.1. Study Population and Procedures

The Health Behaviour in School-Aged Children study (HBSC) is unique in collecting comparable cross-national data on representative samples of young peoples’ health behaviours every fourth year. A standardised international protocol ensures the consistency of measures, sampling, and implementation procedures prepared by the HBSC International Coordinating Centre [46].

Data stem from the HBSC studies conducted in 2006, 2010, 2014, and 2018 in the Czech Republic, Norway, Scotland, and Wales. Data on AST were obtained from a total of 88,212 children across the four time points (2006: *n* = 18,317, 50.3 % girls; 2010: *n* = 18,902, 51.0% girls; 2014: *n* = 17,699, 51.3 % girls; 2018: *n* = 33,294, 50.6 % girls). The age and gender distributions were fairly stable across countries and survey years (Appendix A). Students were surveyed to produce representative national estimates for 11, 13, and 15 year old children. Classes within schools were selected with variations in sampling criteria which allowed us to fit country-level circumstances (e.g., national regions, type of school, and size of schools). Ethical approval for the surveys was obtained at the national level. Participation was voluntary and the children were informed about confidentiality and anonymity. Classroom teachers or trained administrators conducted the survey and consent (explicit or implicit) was given from school administrators and/or parents before participation. More details on the HBSC study procedures can be found elsewhere [46].

### 2.2. Survey Items

#### 2.2.1. Active School Transport

Mode of travel to and from school was assessed with two questions: “On a typical day is the main part of your journey to school made by…?” and “On a typical day is the main part of your journey from school made by…?”. Response options were “Walking”, “Bicycle”, “Bus, train, tram, underground or boat”, “Car, motorcycle or moped” or “Other means”. A slightly different version of the AST items used earlier in the 1985/86 HBSC study have been examined, with the reliability in terms of Cronbach’s alpha found to be 0.83 and a correlation with the total weekday physical activity score—measured by accelerometers—of 0.20 (*p* < 0.01) [47]. In the present study, only 1–1.5% reported “Other means”; this category was included in non-active transport. For the prevalence and trend analyses, AST was both used as a categorical variable based on 4 categories (walking both ways, cycling both ways, active one-way only, non-active transport), and as a categorical variable based on 2 categories (active transport both ways vs. one-way only or non-active transport). The latter categories in both cases were used as a reference in the analysis that follows.

Time to school was assessed with one question “How long does it usually take you to travel to school from your home?” and was used as a proxy for distance to school. Response options were “Less than 5 min”, “5–15 min”, “15–30 min”, 30 min to 1 h” and “More than 1 h”. For the analyses, this variable was recoded into three categories (Less than 5 min, 5–15 min, >15 min). Again, the latter category was used as reference. Travel time to school increased between 2006 and 2018 in Czech Republic and Wales, whereas this remained fairly stable in Norway and Scotland (Appendix A).

#### 2.2.2. Sociodemographic Variables

Gender (boys vs. girls), age groups (11, 13, and 15 year olds), and individual family affluence (FAS—Family Affluence Scale (FAS-II)) were included in the analysis. The latter is a validated HBSC measure of SES [48]. Family affluence is a composite sum score, which resembles a valid measure of household material affluence derived from participants’ responses to 4 items describing the material conditions of their household (respondents’ own household bedrooms, family holidays, family vehicle ownership, and PC ownership). FAS has changed through time but this version was used in 2006, and was therefore applied in the current study. Responses to the individual items are summed on a 9-point scale with set cut-points for low (0 to 3), medium (4 to 5), and high (6 to 9) affluence. The individual FAS responses were combined and standardised by using ridit transformation to give a linear SES-score (0–1). The regression coefficient of the FAS score can be directly interpreted as the predicted difference in AST between the least deprived individual and the most deprived individual. When using this procedure, ordered categorical variables are converted to cumulative probabilities, and the individuals are thus ranked on this continuum. Ridit transformation has previously been applied in inequality studies using SES scales with ordinal measurements [49,50,51] and is recommended for comparisons of the effects of FAS [52]. Family affluence increased between 2006 and 2018 in the Czech Republic, Scotland, and Wales, and remained relatively stable in Norway (Appendix A).

### 2.3. Data Analysis

All analyses were conducted using Stata version 15 (StataCorp LLC, College Station, Texas, USA). Stata’s survey command (svyset) was used to adjust for sampling weight, clustering, and stratification in the sampling design. The alpha level was set to 0.001 given the large sample size and the number of tests. Joint significance of regression terms containing polytomous categorical variables was determined by means of adjusted F-tests. Secular trends were examined both for AST based on 4 categories and for AST based on 2 categories by means of multinomial and logistic regression, respectively. The initial model included age, gender, and country (Section 3.1). In the next step, the country-by-survey year interaction was added (Section 3.2). Pending statistical significance of this interaction term, age- and gender-adjusted results were presented separately for each country. Results were adjusted for age and gender to make sure that changes in AST could not be attributed to changes in age or gender distributions across survey years. Separate analyses were conducted for each survey year, modelled as a categorical variable and as a continuous variable (linear trend). For categorical time, backward difference coding was used to allow for the comparison between consecutive survey years. To determine whether SES and time to school were related to trends in AST, these two variables were added to the model as main effects (Section 3.2). These factors were considered to be potentially explanatory when the OR associated with survey year was reduced by ≥10% [53]. For ease of interpretation, the remaining models were only conducted for AST based on 2 categories and with survey year modelled as a categorical variable (Section 3.3). Country differences were explored further by adding the two-way interactions of country by gender, age, SES, and time to school, respectively. Finally, potential country differences over time were explored by testing a model with three-way interactions of country by survey year by, respectively, gender, age, SES, and time taken to travel to school. For statistically significant interaction terms, country differences were examined across survey years.

## 3. Results

### 3.1. AST by Country across Surveys

For AST based on four categories, the adjusted F-test for country was F_(9, 2258)_ = 204.5, *p* < 0.001, and for AST based on two categories F_(3, 2264)_ = 136.1, *p* < 0.001. This indicated significant variation in AST by country for all survey waves combined (Table 1). Walking to school both ways was most common in the Czech Republic (55%) and least common in Wales (30%), whereas cycling to school was limited, with the exception of Norway (22%). One-way AST was relatively uncommon and there was also only modest variation in its prevalence between countries (6–10%). When considering active travel both ways (Table 1) the prevalence in the Czech Republic (57%) and Norway (59.4%) was significantly higher as compared to Scotland (46%) and Wales (31%).

### 3.2. Secular Trends in AST by Country

The country-by-categorical survey year interaction was statistically significant for both AST based on four categories (F_(9, 2258)_ = 204.5, *p* < 0.001) and AST based on two categories (F_(9, 2258)_ = 204.5, *p* < 0.001), indicating that the effect of survey year varied across countries. Similar results were obtained for treating time as a continuous variable. As shown in Table 2, AST changed significantly over time in Czech Republic, but remained stable in the other three countries. In the Czech Republic, there was a relatively sharp decrease in walking both ways between 2006 and 2010, followed by a stable pattern between 2010 and 2018. A similar, though less pronounced, pattern was found for cycling both ways. A small linear increase over time was observed for one-way AST in the Czech Republic. For AST based on two categories, this translated into a decrease in AST both ways between 2006 and 2010, followed by a stable pattern between 2010 and 2018. Despite the overall decrease in the Czech Republic between 2006 and 2018, the prevalence of AST both ways remained higher when compared to Scotland and Wales (Figure 1).

Adding family affluence and time to school to the basic country-specific models (age, gender, survey year) did not change the effect of survey year in Norway, Scotland, and Wales. In the Czech Republic, the odds ratio representing the change from 2006 to 2010 in walking both ways (based on four categories) changed from 0.57 to 0.64 after adding family affluence, 0.59 after adding time to school, and 0.65 after adding both variables. This equates to an OR-change of, respectively, 16, 5, and 19%. Family affluence and time to school did not change the significant effect of survey year on cycling both ways, neither did these variables change the linear effect of survey year on one-way AST. For AST based on two categories, the odds ratio representing a change from 2006 to 2010 in AST both ways changed from 0.54 to 0.60 after adding family affluence, 0.56 after adding time to school, and 0.62 after adding both variables. This equates to OR-change of 13, 4, and 17%, respectively. Overall, these results indicated that change over time in family affluence was associated with a change in walking both ways in the Czech Republic. It should be noted that all the mentioned effects of survey year in the Czech Republic remained statistically significant after adding family affluence and time to school to the model.

### 3.3. Country Differences in AST Both Ways by Gender, Age Group, Family Affluence and Time to School

All four two-way interactions of gender, age group, family affluence, and time to school by country were statistically significant (*p* < 0.001), indicating that the effects of these variables varied by country. As shown in both Table 3 and Figure 2, there were significant gender differences in Scotland and Wales, with boys being more likely to exhibit AST both ways as compared to girls in these two countries. There was a particularly strong age group effect in Norway with 11 year olds being much more likely to exhibit AST both ways as compared to 13 and 15 year olds. Family affluence did not impact the probability of AST in Norway, whereas children in the other countries from more affluent families were less likely to have AST both ways as compared to their counterparts from less affluent families. The estimated probability difference of AST between children coming from the least affluent families (0) and children coming from the most affluent families (1) is displayed in Figure 2. Finally, the effect of time to school on AST was much stronger in the Czech Republic when compared with the other three countries. Children from the Czech Republic with a travel time of less than 5 min were more likely to engage in AST (86%), whereas this was much less likely in Scotland (62%) and Wales (50%). This was also true for travel times between 5–15 min, but the country differences were somewhat less pronounced for this category. Across surveys, the prevalence of travel time less than 5 min was 21% for the Czech Republic, 17% for Norway, 22% for Scotland, and 14% for Wales, whereas the prevalence of travel time between 5–15 min was 48% for the Czech Republic, 46% for Norway, 44% for Scotland, and 44% for Wales.

None of the four three-way interactions of country by survey year by, respectively, gender, age, SES, and time to school were statistically significant, indicating that the observed country differences did not change over time.

## 4. Discussion

The current study provides recent trends in AST and their associations with gender, age, SES, and time to school across four European countries. The findings demonstrate that, apart from the Czech Republic, there were generally stable patterns of partly low levels of AST in the period from 2006 to 2018. However, the prevalence and association with moderating factors varied considerably between countries. For all survey waves combined, walking to school both ways were most common in the Czech Republic (55%), followed by Scotland (45%), Norway (37%), and Wales (30%). Cycling to school was only common in Norway, with the highest prevalence of total AST (59%). One-way AST was relatively uncommon with modest variation between countries. Overall, the results show that active travel both ways was considerably higher in the Czech Republic and Norway when compared to Scotland and Wales. Although, there was a general direction towards a decline across all four countries, this was non-significant. The stable trends in AST observed in the current study contrast with most studies [19,20,21,22,23,24,25,26,27,28] but are in line with one study from Australia that assessed trends between 2004–2010 [54] and the findings from 28 studies in Spain between 2010 to 2017 [41]. The findings suggest that a general decline in AST may have levelled out in various regions in Europe.

The prevalence of AST changed significantly over time only in the Czech Republic, with a relatively sharp decrease observed between 2006 and 2010, followed by a stable pattern between 2010 and 2018. This is in line with a previous study [42]. A small linear increase over time was also observed for one-way AST in the Czech Republic. The extent to which the recent initiation of services, such as bike shares, has contributed to this increase is unknown, but a topic worth further investigation. The market demand for micro-mobility is also expected to grow significantly. The availability of these personal vehicles may also influence AST in the future. The negative trend observed in the Czech Republic might be explained by increasing car ownership [55], insufficient cycling infrastructure at a municipal and school level [56], and barriers dealing with safety concerns [57]. The Czech Republic was also the only country where family affluence was associated with change over time in walking both ways. It may, therefore, be the case that the change in family affluence contributed to some of the change in AST, for example, through increased availability of family cars. In all countries, except Norway, children from more affluent families were less likely to engage in AST both ways when compared to their counterparts from less affluent families. This finding is in line with studies from North America and New-Zealand [14,36]. In addition to the possibility of reduced access to a private car in lower-income families, constraints on the time available for single-parents to drive has been identified as a potential reason for family affluence differences [36].

The travel time to school increased between 2006 and 2018 in the Czech Republic and Wales, whereas this remained relatively stable in Norway and Scotland. School distance and time to school have been identified as strong correlates of AST [6,14,29,36,58]. It has been suggested that the development of bigger school units and the closing of neighbourhood schools and school choice policies are contributors to the increased distance and time to school [29,36]. Nevertheless, the literature suggests that many children live within a reasonable walking distance from school [59,60,61]. In the current study, the prevalence of a travel time less than 15 min was relatively similar across countries. An interesting finding was that the effect of time to school on AST was strongest in the Czech Republic, as Czech children with a travel time of less than 5 min were much more likely to engage in AST, as compared to children in Scotland and Wales. This was also observed for travel times between 5–15 min, but the country differences were less pronounced. Thus, there seems to be potential, especially for countries such as Scotland and Wales, to increase their levels of AST by targeting families who live within a “threshold” distance. This is reported to be less than 1.5–2.0 km for walking and 3.0 km for biking [62,63,64]. However, environmental factors (e.g., how many busy roads to cross, presence of staff to help crossing roads) could also be influential in short distances. The findings highlight the need for comprehensive local approaches, which should also include some form of risk assessment and environmental modifications.

The relatively high and stable levels of AST observed in Norway may partly be a result of a substantial proportion of schoolchildren cycling to school. Cycling to school allows students to move faster and they can cover greater distances [65]. Unlike studies from many other European countries, schoolchildren from Scotland, Wales and the Czech Republic did not use cycling as a mode of travel. However, the findings are in line with studies from Ireland [66]. Higher rates of cycling versus walking to school among adolescents have been found in other Nordic countries, such as Denmark and Finland [29]. In contrast to Denmark, known for its longstanding cycling traditions, cycling-friendly infrastructure, and flat landscape, the climate, topography, and physical environmental conditions in Norway are diverse. The relatively high prevalence of cycling could be related to social and cultural norms. Nordic countries have a culture where outdoor activities, in general, play an important role in the “way of life” [67] and Norway has a long tradition for outdoor education in primary schools [68]. This could have a positive impact on attitudes towards independent mobility, which has been associated with cycling to school [69]. Another, but perhaps related issue is expectations around school dress codes. In New Zealand, school uniform requirements have been found to influence adolescents’ motivation for cycling to school [70]. This could also be the case in Scotland and Wales, where school uniforms can be quite formal, whereas in Norway students do not use school uniforms. In a study from Scotland, wearing helmets were also a barrier for cycling to school, especially among older children [71]. Other essential factors that could explain why school children do not cycle to school, include lack of cycling infrastructure (e.g., dedicated cycle paths), limited facilities at school and children’s lack of competence in terms of cycling [66]. The reality is most likely a combination of all the factors noted.

In the current study, an age effect was observed, with lower odds for AST for the 13 and 15 year olds in Norway and the 13 year olds in Scotland. In Wales 11 year olds are already at secondary school which would explain the lack of age effect. In terms of the secular trends in AST, we did not observe any age interactions. The effect of age was considerable in Norway. This is in line with previous studies that have found substantially higher levels of AST among primary as compared to secondary school children [11,47]. Secondary schools are typically bigger school units with increased travel distances for most students [14], which might explain this finding.

Combining the data from all study waves, it appears that gender was a significant factor influencing AST in Wales and Scotland. Gender differences have been confirmed in several previous studies [14,32,36,39,40]. In terms of secular trends in AST, we did not observe any gender interactions, suggesting that the gender differences in Wales and Scotland remained constant throughout the years. Findings from other trend studies have shown comparable results [25,34,41,72]. In two studies from Spain and Brazil, no gender differences were observed [27,43]. However, over time there was a decline for Spanish girls [43], and a decline for boys and an increase for girls among Brazilian youth [27]. Suggested explanations for girls showing lower AST levels relate mainly to safety concerns (e.g., traffic and crime) and independent mobility, with boys more likely to be allowed to explore their neighbourhood environment to a greater extent without supervision [10]. It has also been suggested that the current generation of parents have more concerns about safety that might be responsible for some of the differences in AST levels [31]. Nevertheless, the differential gender differences observed across countries, suggests cultural variations that future studies should seek to understand more in depth.

### Strengths and Limitations

The key strengths of this study include the large sample size and ability to compare across four European countries, with representative samples of adolescents, using a shared research protocol. Furthermore, the use of standard methodology for data collection and measurement of AST, with consistent wording throughout the waves, ensured internationally comparable data and robust trend analyses. The analyses were conducted with rigour, by adjusting for sampling weight, clustering and stratification in the sampling design, and the alpha level was set to 0.001, given the large sample size and the number of tests. The possibility of differentiating between cycling and walking in this study was also important, as it gave a more detailed picture of students’ travel behaviours to school.

However, there are also limitations. The analysis is based on a repeated cross-sectional design conducted every fourth year. Interpretation of trends should be made carefully because there is no information available between the different survey waves. Data collection was conducted by self-report, and therefore, may be susceptible to recall bias. However, it is a challenge to objectively assess AST because of its versatile nature [73]. The AST measure does not detect day-to-day variations in transport behaviour and the students were categorised based on their main method of travel, which could misrepresent the amount of activity. For example, respondents who used passive transport may have accumulated some amount of physical activity for a segment of the journeys, that could add to their daily physical activity uptake. Furthermore, distance to school was approximated by the included time to school variable, which was a suboptimal solution as time to school partly dependents on transportation mode and could also be influenced by time spent stopping somewhere during the journey and other environmental factors. Nevertheless, a strength of HBSC lies in its breadth, which as a consequence, unfortunately means that not all issues can be explored in great depth.

## 5. Conclusions

The study found stable patterns of AST in the period from 2006 to 2018, except for a reduction in the Czech Republic from the first to the second wave. These findings could indicate that the previously observed decrease in AST has been flattening off in the countries studied here. Still, the findings suggest that there is a great potential to increase the level of active commuting to school, especially in Scotland and Wales where the levels were low despite government action. This indicates that tackling active travel alone (e.g., with a focus on infrastructure) is not enough and points to the fact that action really does need to be cross-cutting and comprehensive. The variation in the prevalence of AST and the observed associations with gender, age, family affluence and time to school, suggest there are most likely country-specific factors influencing students’ choice of travel mode to school.

## Figures and Tables

**Figure 1 ijerph-18-02118-f001:**
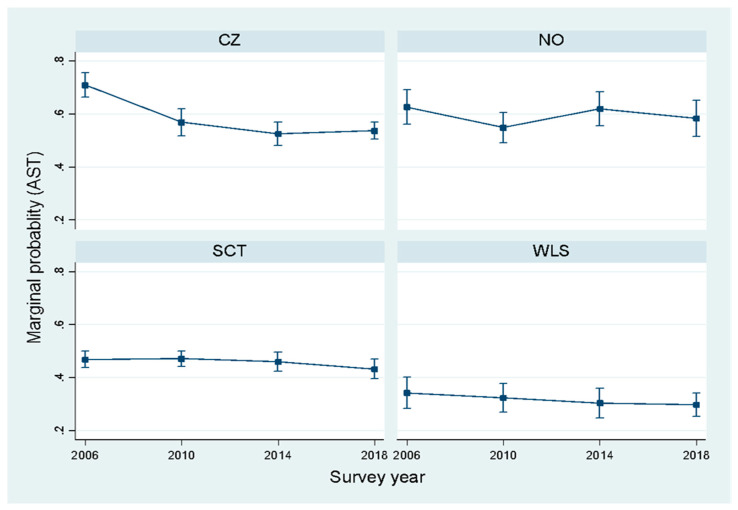
Age and gender adjusted prevalence of AST both ways by country and survey year. (CZ = Czech Republic, NO = Norway, SCT = Scotland, WLS = Wales).

**Figure 2 ijerph-18-02118-f002:**
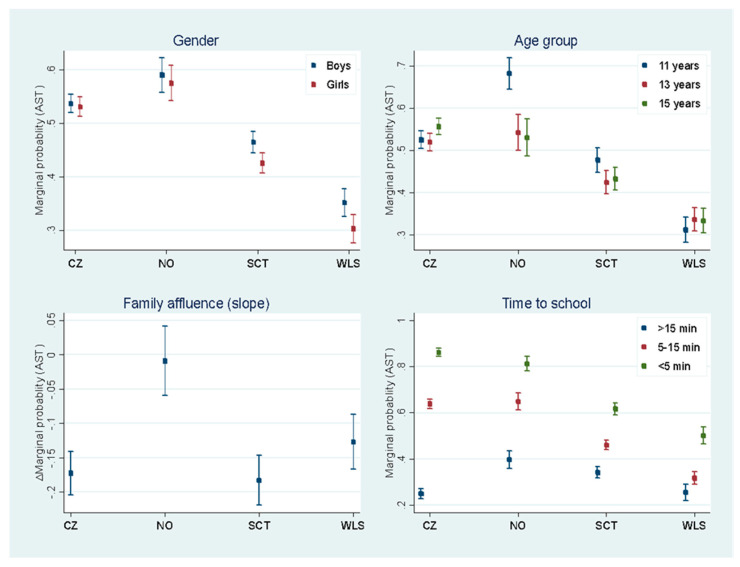
Country differences in AST both ways by gender, age, family affluence, and time to school.

**Table 1 ijerph-18-02118-t001:** Age/gender-adjusted prevalence of active school transport (AST) by country across survey years.

AST Based on 4 cat.	Country	% (99.9% CI)	OR (99.9% CI)
Walking both ways	Czech Republic	54.6 (51.9, 57.3)	**1.53 (1.21, 1.93)**
	Scotland	44.6 (42.5, 46.7)	0.95 (0.76, 1.19)
	Wales	30.1 (26.5, 33.7)	**0.47 (0.36, 0.62)**
	Norway	36.6 (33.0, 40.3)	Ref.
Cycling both ways	Czech Republic	2.4 (1.7, 3.2)	**0.11 (0.07, 0.16)**
	Scotland	1.3 (.08, 1.7)	**0.04 (0.03, 0.06)**
	Wales	0.8 (.04, 1.2)	**0.02 (0.01, 0.03)**
	Norway	22.2 (19.2, 25.2)	Ref.
One-way AST	Czech Republic	8.7 (7.8, 9.6)	**1.53 (1.21, 1.94)**
	Scotland	9.6 (8.7, 10.4)	**1.29 (1.02, 1.62)**
	Wales	8.6 (7.5, 9.8)	0.85 (0.65, 1.12)
	Norway	5.9 (5.0, 6.7)	Ref.
**AST based on 2 cat.**			
AST both ways	Czech Republic	57.2 (54.5, 59.8)	0.91 (.74, 1.12)
	Scotland	46.0 (43.9, 48.1)	**0.58 (0.48, 0.70)**
	Wales	31.0 (27.3, 34.6)	**0.30 (0.24, 0.39)**
	Norway	59.4 (55.3, 63.6)	*Ref.*

Ref. active travel 4 cat. = no AST based on multinominal logistic regression, ref. active travel 2 cat. = no AST/one-way AST. Estimates in bold = *p* < 0.001, logistic regression.

**Table 2 ijerph-18-02118-t002:** Age/gender-adjusted prevalence and secular trends in AST by country and survey year *.

AST Based on 4 cat.	Country	2006 (%)	2010 (%)	2014 (%)	2018 (%)	2010 vs. 2006 OR (99.9%CI)	2014 vs. 2010 OR (99.9%CI)	2018 vs. 2014 OR (99.9%CI)	Adj. F-Test (*p*-Value)	Linear trend OR (99.9%CI)
Walking both ways	Czech Republic	67.0	54.9	50.0	51.5	**0.57 (0.37, 0.87)**	0.85 (0.58, 1.27)	1.11 (0.81, 1.51)	**<0.001**	**0.96 (0.94, 0.99)**
	Norway	41.9	32.1	37.5	34.1	0.61 (0.35, 1.07)	1.43 (0.83, 2.45)	0.77 (0.43, 1.37)	0.01	0.98 (0.93, 1.02)
	Scotland	45.7	46.0	43.9	42.1	0.99 (0.77, 1.27)	0.92 (0.70, 1.21)	0.90 (0.66, 1.21)	0.08	0.98 (0.96, 1.01)
	Wales	33.5	31.6	29.9	28.8	0.94 (0.56, 1.58)	0.91 (0.54, 1.54)	0.92 (0.56, 1.49)	0.34	0.98 (0.94, 1.02)
Cycling both ways	Czech Republic	3.8	1.8	2.3	2.2	**0.33 (0.13, 0.85)**	1.20 (0.43, 3.30)	1.02 (0.40, 2.61)	**<0.001**	0.94 (0.88, 1.02)
	Norway	20.2	22.0	24.1	23.5	0.81 (0.43, 1.51)	1.38 (0.78, 2.43)	0.85 (0.44, 1.65)	0.30	1.00 (0.95, 1.06)
	Scotland	1.0	1.0	2.0	1.1	0.90 (0.40, 2.04)	1.88 (0.81, 4.40)	0.55 (0.22, 1.35)	0.06	1.02 (0.96, 1.08)
	Wales	0.7	0.7	0.5	0.9	0.92 (0.29, 2.93)	0.73 (0.15, 3.48)	1.68 (0.37, 7.55)	0.70	1.02 (0.93, 1.12)
One-way AST	Czech Republic	4.1	7.3	9.5	10.8	1.25 (0.77, 2.06)	1.23 (0.80, 1.88)	1.21 (0.87, 1.68)	**<0.001**	**1.05 (1.02, 1.08)**
	Norway	5.1	6.1	5.8	6.9	0.94 (0.55, 1.62)	1.22 (0.71, 2.08)	1.02 (0.56, 1.85)	0.48	1.01 (0.97, 1.06)
	Scotland	10.0	9.5	9.2	9.6	0.94 (0.69, 1.27)	0.94 (0.68, 1.30)	0.98 (0.70, 1.36)	0.40	0.99 (0.96, 1.01)
	Wales	8.9	10.8	10.4	7.3	1.20 (0.68, 2.13)	0.93 (0.52, 1.65)	0.67 (0.40, 1.10)	0.005	0.97 (0.93, 1.00)
**AST based on 2 cat.**										
AST both ways	Czech Republic	71.0	56.9	52.6	53.7	**0.54 (0.36, 0.80)**	0.84 (0.59, 1.20)	1.05 (0.79, 1.40)	**<0.001**	**0.95 (0.93, 0.97)**
	Norway	62.7	54.9	62.0	58.4	0.68 (0.42, 1.10)	1.36 (0.87, 2.12)	0.80 (0.48, 1.31)	0.02	0.98 (0.94, 1.03)
	Scotland	46.8	47.2	46.0	43.3	1.01 (0.80, 1.26)	0.95 (0.75, 1.21)	0.88 (0.67, 1.16)	0.11	0.99 (0.97, 1.01)
	Wales	34.3	32.4	30.4	29.8	0.91 (0.57, 1.45)	0.92 (0.57, 1.46)	0.98 (0.63, 1.52)	0.44	0.98 (0.95, 1.02)

* Ref. AST 4 cat. = no AST, ref. AST 2 cat. = no AST/one-way AST. Estimates in bold = *p* < 0.001. The reported results of the adjusted F-tests and linear trends for the 4-category active travel variable are within country and within the relevant outcome category. The joint significance of survey year across outcome categories was only statistically significant for Czech Republic (*p* < 0.001).

**Table 3 ijerph-18-02118-t003:** Adjusted associations with AST both ways by country *.

Variables	Czech Republic	Norway	Scotland	Wales	
	OR (99.9%CI)	OR (99.9%CI)	OR (99.9%CI)	OR (99.9%CI)	Notable Country Differences
**Survey year**					
2010 vs. 2006	**0.61 (0.43, 0.89)**	0.67 (0.41, 1.08)	1.11 (0.88, 1.39)	0.96 (0.61, 1.52)	Change in AST in CZ only
2014 vs. 2010	0.99 (0.72, 1.36)	1.35 (0.87, 2.10)	0.96 (0.76, 1.22)	0.99 (0.63, 1.55)	-
2018 vs. 2014	1.01 (.78, 1.31)	0.83 (0.51, 1.34)	0.91 (0.70, 1.19)	1.06 (0.70, 1.60)	-
**Female gender**	0.97 (0.87, 1.08)	0.93 (0.81, 1.07)	**0.85 (0.76, 0.94)**	**0.79 (0.70, 0.90)**	Gender differences in SCT, WLS only
**Age group**					
13 year olds	0.97 (0.83, 1.14)	**0.51 (0.37, 0.71)**	**0.80 (0.64, 0.99)**	1.13 (0.97, 1.31)	Age group effect in NO, SCT only
15 year olds	1.18 (1.00, 1.40)	**0.49 (0.36, 0.67)**	0.83 (0.67, 1.03)	1.11 (0.92, 1.35)	Age group effect in NO only
**Time to school**					
<5 min.	**20.10 (15.81, 25.56)**	**6.88 (5.23, 9.05)**	**3.15 (2.62, 3.79)**	**2.97 (2.20, 4.01)**	Stronger time to school effects in CZ, NO
5–15 min.	**5.58 (4.81, 6.57)**	**2.90 (2.41, 3.50)**	**1.65 (1.43, 1.90)**	**1.36 (1.09, 1.69)**	Stronger time to school effects in CZ, NO
**Family affluence**	**0.40 (0.32, 0.50)**	0.96 (0.70, 1.31)	**0.46 (0.37, 0.56)**	**0.55 (0.43, 0.70)**	Social gradient absent in NO only

* Estimates by country for model AST = Survey year + gender + Age group + Time to school + Family affluence. Ref. AST 2 cat. = no AST/one-way AST. CZ = Czech Republic, NO = Norway, SCT = Scotland, WLS = Wales. Ref. Age group = 11 year olds, Ref. Time to school = >15 min. Family affluence ridit transformed to a linear score (0–1). Estimates in bold = *p* < 0.001.

## Data Availability

The University of Bergen is the data-bank manager for the HBSC study. Please contact the corresponding author for data requests.

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
