# Peer review of "12-Year Trends in Active School Transport across Four European Countries—Findings from the Health Behaviour in School-Aged Children (HBSC) Study"

_ijerph, 2021, doi:10.3390/ijerph18042118_

Round 1

Reviewer 1 Report

The introduction gives a thorough background of AST and positions the present research in the domain.

The analysis is well presented, however there are some points that would need further explanation/elaboration: 

Travel time is related to the mode used for the trip. Different mode preferences prevail in each country and year. Please explain how this time is comparable among subjects and countries.

Please provide some explanation on the age/gender adjustment.

Line 208 comes in rather abruptly. It is not clear how parameters are used.  What is the basic model? What are its modifications? Please use a table, to show the different forms of the model and the results, described from line 208 to 222. What is the actual meaning of these changes in the OR values (e.g. form .57 to .59 etc.), in regards to AST?

Author Response

REVIEWER 1:

The introduction gives a thorough background of AST and positions the present research in the domain.

The analysis is well presented, however there are some points that would need further explanation/elaboration: 

  1. Travel time is related to the mode used for the trip. Different mode preferences prevail in each country and year. Please explain how this time is comparable among subjects and countries.

Response: Travel time was used as a proxy for distance to school as described in section 2.2.1, but this was suboptimal as rightly pointed out by the reviewer. Unfortunately, we do not have other data available to examine its validity more thoroughly. We included this as a limitation in the discussion section, see lines 382-383.

  1. Please provide some explanation on the age/gender adjustment.

Response: The following line was added to the analysis section on page 4, line 174-176:

“Pending statistical significance of this interaction term, age- and gender-adjusted results were presented separately for each country. Results were adjusted for age- and gender to make sure that changes in AST could not be attributed to changes in age- and gender distributions across survey years. Separate analyses were conducted for survey year modelled as a categorical variable and as a continuous variable (linear trend).”

  1. Line 208 comes in rather abruptly. It is not clear how parameters are used.  What is the basic model? What are its modifications? Please use a table, to show the different forms of the model and the results, described from line 208 to 222. What is the actual meaning of these changes in the OR values (e.g. form .57 to .59 etc.), in regards to AST?

Response: Our modeling approach is described in the “data analysis” section and we have now added information about the relevance of the different analysis stages for the different sections in the results (line 172-184).

All analyses were conducted using STATA version 15. Stata's survey command (svyset) was used to adjust for sampling weight, clustering, and stratification in the sam-pling design. Alpha level was set to .001 given the large sample size and the number of tests. Joint significance of regression terms containing polytomous categorical variables was determined by means of adjusted F-tests. Secular trends were examined both for AST based on 4 categories and for AST based on 2 categories by means of multinomial and lo-gistic regression, respectively. The initial model included age, gender, and country (section 3.1). In the next step, the country-by-survey year interaction was added (section 3.2). Pending statistical significance of this interaction term, age- and gender-adjusted results were presented separately for each country. Separate analyses were conducted for survey year modelled as a categorical variable and as a continuous variable (linear trend). For categorical time, backward difference coding was used to allow for the comparison be-tween consecutive survey years. To determine whether SES and time to school were related to trends in AST, these two variables were added to the model as main effects (section 3.2). These factors were considered to be potentially explanatory when the OR associated with survey year was reduced by ≥10% [47]. To ease interpretation, the remaining models were only conducted for AST based on 2 categories and with survey year modelled as a categorical variable (section 3.3). Country differences were explored further by adding the two-way interactions of country by gender, age, SES, and time to school, respectively. Finally, potential country differences over time were explored by testing a model with the three-way interactions of country by survey year by, respectively, gender, age, SES, and time taken to get to school. For statistically significant interaction terms, country differences were examined across survey years.

Moreover, line 217-218 was changed to clarify model changes:

“Adding family affluence and time to school to the basic country-specific models (age, gender, survey year) did not change the effect of survey year in Norway, Scotland, and Wales.”

How to interpret changes in OR was also described in the data analysis section:

These factors were considered to be potentially explanatory when the OR associated with survey year was reduced by ≥10% [47].

We are hesitant to add an extra table because the results in lines 208-222 are mostly relevant for Czech Republic only, and because the manuscript already contains three tables and two figures. We hope therefore that the provided clarifications in the text are sufficient for the reviewer, but please let us know if not.

See track changes in revised manuscript.

Reviewer 2 Report

The article is very interesting and addresses an issue of vital importance to the population. The introduction is quite well structured and well defined, although the bibliographic references are current, one of the elements to improve this section would be to add papers from 2019 and 2020. The objectives are well designed and the material and methods satisfactorily address the article, The results are well exposed as well as discussion, conclusions and limitations. The paper is ok for publication and its correction has been very satisfactory since it does not present difficulties. Congratulations to the authors

Reviewer 3 Report

This paper analyses the trends of active transport among school children across four countries in the Europe, and differences between the socio-demographic and economic classifications. A few minor suggestions:

  1. Line 101- HBSC in full.
  2. Table 1- re-align Norway to the first or last row for comparison 
  3. line 194- could the significant effect of survey year varied across countries be due to the implementations of different policies in different years by the different countries?
  4. line 212- could you show how did you calculate the percent OR-change of 16%, 5%, 19%?
  5. Table 3-  I suppose the reference here is 11 year old, and >15 min for the respective countries for age group and time to school?
  6. line 272- is there any recent initiation of service that cause this increasing trend of one way AST? such as bike share?
  7. line 339- 'decline for girls, boys, girls...' this is in the introduction too? possible to reframe this statement?
  8. line 366- could you elaborate more on the 'meaningful PA'?
  9. line 368- travel time- possibly also included time stopping-by somewhere during the journey to/from home which extended the journey time - the environment factors?
